# Multi-F-Structured MOF Materials Enhance Nanogenerator Output Performance for Corrosion Protection of Metallic Materials

**DOI:** 10.3390/molecules28237894

**Published:** 2023-12-01

**Authors:** Jia-Bin Xiong, Yong-Juan Zhou, Shi-Hui Wang, Zhang-Qi Xiong, Zi-Kun Zhang, Shan-Shan Zhang, Chen-Kunlun Zhang, Chao-Fan Xu, Guo-Qun Liu

**Affiliations:** School of Material and Chemical Engineering, Center for Advanced Materials Research, Zhongyuan University of Technology, Zhengzhou 450007, China; zhouyongjuan02@foxmail.com (Y.-J.Z.); wangshihui1115@foxmail.com (S.-H.W.); xiongzhangqi@foxmail.com (Z.-Q.X.); zhangzikun5137@foxmail.com (Z.-K.Z.); zss-630x@foxmail.com (S.-S.Z.); zhangchenkl@foxmail.com (C.-K.Z.); xuchaofan0127@foxmail.com (C.-F.X.)

**Keywords:** TENG, MOF, corrosion protection, cathodic protection

## Abstract

MOF (metal organic framework) materials have been used as functional materials in a number of fields due to their diverse spatial tunability, which produces rich porous structures with stable and continuous pores and a high specific surface area. A triboelectric nanogenerator can convert trace mechanical energy into electrical energy, and the application of MOF materials to triboelectric nanogenerators has been intensively studied. In this work, we report on two MOFs with similar spatial structures, and the modulation of the end microstructures was achieved using the difference in F content. The output performance of friction power generation increases with the increase in F content, and the obtained polyacidic ligand materials can be used to construct self-powered corrosion protection systems, which can effectively protect metallic materials from corrosion.

## 1. Introduction

With the development of society, the gradual depletion of fossil energy sources and the increase in environmental pollution have forced us to accelerate the search for green and sustainable energy sources. Nature provides ample sustainable energy sources, such as solar, thermal, wind, water, and mechanical energy, which, if utilized, can partially mitigate the depletion of fossil fuels and the energy crisis [1,2,3,4,5]. However, solar, wind, and other sustainable energy sources are costly to collect and construct, are vulnerable to climate change, among other factors, which prevents them from being popularized on a large scale, and their impacts are geographically limited [6]. First invented by Zhonglin Wang and his team in 2012 [7,8,9,10,11], the friction nanogenerator (TENG) aims to convert tiny quantities of mechanical energy into electrical energy by utilizing the coupling of the frictional electrification effect and the electrostatic induction effect. Friction electric nanogenerators (TENGs) have been favored in recent years as a new type of energy storage and output device that directly recovers almost all types of mechanical energy present in our daily lives and in nature. Compared to other energy sources, harvesting mechanical energy is easier because it is commonly found in nature and in people’s daily lives. For example, the flow of air and water, the movement of the human body, and even the slightest change in pressure somewhere in the body due to breathing, heartbeat, or blood flow, all have the potential to drive nanogenerators to generate electricity. Therefore, the TENG can be regarded as a sustainable renewable energy source [12,13,14]. The merit of TENGs’ output performance is closely related to the charge density of TENG electrode materials. Increasing the charge density of the contact layer can improve the performance of TENG, and by creating curved surfaces it can improve electric charge density (σ) and show the excellent output performance of nanocomposites with structural, polarization, surface functionalization, doping, and other treatments and applications [15]. One of the effective ways to improve the output properties of the TENG is to increase the effective contact area by preparing a micro/nano weave on the contact surface [16,17].

Metal organic frameworks (MOFs), consisting of metal ions/clusters and organic linkers, are a class of highly crystalline porous materials with a large specific surface area, high porosity, tunable structure, and a rich variety of crystalline materials and other characteristics [18,19]. With diverse structures and highly tunable pore sizes, MOFs are able to cover the complete pore size gap between microporous zeolites and mesoporous silica. With a large number of metal nodes and a theoretically infinite number of organic linkers, the composition and structure of MOFs can be easily tuned to precisely realize the target function. The superior properties of MOFs have led to their widespread use in the study of gas storage [20,21], molecular separations [22,23], catalysis [24,25], chemical sensing [26,27,28], and bioimaging [29,30,31,32]. The output performance of TENGs is mainly determined by the nature of frictionally electroactive materials, and MOFs provide flexibility in the size, function, and geometry of TENGs. The easily tunable surface of MOFs in TENGs and the friction layer with a wide specific surface area increase the contact area and charge output of TENGs.

The influence of the overall negative electric effect of the negative electronegative groups on the output performance of friction power generation is investigated while keeping the spatial structure approximately the same. Exploring the conformational relationship, the effect of terminal microstructure on TENGs was thoroughly investigated to provide a theoretical basis for improving the output performance of TENGs based on a metal–organic skeleton.

In this paper, we choose to synthesize MOF materials from the same Co metal center and different ligands to use as friction power generation materials. The MOF material synthesized from 2,3,5,6-tetrafluoroterephthalic acid ligand is called compound **1**, and the MOF material synthesized from 2,5-bis(trifluoromethyl)terephthalic acid ligand is called compound **2**. The Co-TENG devices prepared from the corresponding electrode materials were named **1**- and **2**-TENG. The effect of the number of F groups on the output properties of TENGs was investigated by controlling the body structure to change the terminal structure. The electrochemical characterization test results show that compound **2**-TENG has the best output performance, followed by **1**-TENG; compound **2** has more electronegative groups, and electronegative groups can improve the output performance of TENGs. In order to protect metal materials from corrosion, we introduce it into the self-powered cathodic protection system, and the TENG-based cathodic protection system realizes the ideal model of utilizing nature’s energy to protect metals from corrosion. The self-powered cathodic protection system has the potential to be used in daily life, industrial production, and marine development due to its advantages of low cost, no external energy consumption, and ease of preparation.

## 2. Results

Compound **2** was synthesized using a hydrothermal method centered on the metal Co and 2,5-bis(trifluoromethyl)terephthalic acid. The crystal structure of the compound is shown in Figure 1. In compound **2**, the central metal Co ion is in the geometrical configuration of a six-coordinated octahedron, in which the four oxygen atoms are derived from the carboxyl groups in each of the two ligands, and the two oxygen ligands come from water molecules, and the organic ligand uses all of its oxygen atoms to form a ligand bond with the metal-centered cobalt (Figure 1a). The coordination mode of the organic ligand of compound **2** is that one ligand connects to four metal-centered cobalt ions, the four cobalt ions all connected to oxygen atoms in the carboxyl group on the ligand (Figure 1b), and the compounds are three-dimensional mesh structures consisting of chains of octahedral Co atoms in a coordination environment of O_6_ (Figure 1c). Figure 1d is a plan view of the compound shown along the a-axis direction and Figure 1e shows the three-dimensional mesh structure of the compound; it is clear from Figure 1d,e that the compound is a rhombic coordination polymer.

Compounds **1** and **2** were characterized and tested via XRD, FT-IR, and XPS. The XRD patterns of the compounds were in agreement with the previously reported PXRD patterns (Figure 2a) and both fitted well, indicating the good crystallinity and purity of the compounds. The IR spectra of the compounds showed that the peaks of IR at 1550–1650 cm^−1^ proved the presence of the C=O bond and the peaks at 1100–1350 cm^−1^ and 700–1100 cm^−1^ proved the presence of the C-F bond and the C-C bond (Appendix A and Figure 2b). The valence and elemental composition of Co in the compounds were analyzed using the profiles obtained from the XPS tests of the compounds, and the full XPS profile of compound **2** illustrated that it contained the elements Co, F, C, and O (Figure 2c). The peaks at 781.99 eV for compound **1** corresponded to those for Co 2p_3/2_ and those at 797.78 eV corresponded to those for Co 2p_1/2_ (Appendix A); the peaks at 781.78 eV and 797.51 eV for compound **2** corresponded to those for Co^2+^ 2p_3/2_ and 2p_1/2_, respectively (Figure 2d), both confirming the presence of Co in the form of Co^2+^.

Appendix A shows the thermogravimetric analysis (TGA) curves and N_2_ adsorption-desorption isotherms of compounds **1** and **2** after drying in an oven at 70 degrees Celsius for 4 h. Compounds **1** and **2** show significant weight loss at 110 and 223 °C Celsius as a result of the volatilization of residual DMF from the compounds. The BET specific surface areas of compounds **1** and **2** were 10.25 m^2^/g and 33.00 m^2^/g, and the adsorption-desorption values were 19.81 cm^3^/g and 61.54 cm^3^/g, respectively, which were type IV adsorption–desorption isotherms, and the pore size distribution curves indicated the presence of mesopores in the materials.

Prior to the TENG output performance test, we performed the Mott—Schottky test and the UV-2600 spectroscopic test for compounds **1** and **2** (Figure 3). The test data allowed us to determine the semiconductor type of the compounds as well as the highest occupied orbital (HOMO) and the lowest unoccupied molecular orbital (LUMO), and to prejudge the output performance of the TENG. Compound **1** has UV peaks at 325 and 550, and compound **2** has UV peaks at 280 and 550. Both materials have π→π* and n→π* absorption bands, with both double-bonded and carbonyl n-electronic absorption in the aromatic ring. With respect to Compound **1**, because of the electron-rich structure, the peak is shifted to the right and the band gap is reduced to enhance conductivity. From the UV-Vis diffuse reflectance spectra of compounds **1** and **2**, the bandgap energies (Eg) of compounds **1** and **2** can be seen to be about 1.61 eV and 3.98 eV, respectively (Figure 3b,d). Compounds **1** and **2** are both n-type semiconductors from the positive slopes of the Mott—Schottky curves versus the potential curves (Figure 3e,f). The lowest unoccupied molecular orbital (LUMO) flat bands of compounds **1** and **2** are −0.57 V and −0.64 V, respectively, when Ag/AgCl is used as a reference (obtained at the union of −0.77 V and −0.84 V, respectively), and the HOMO orbitals of **1** and **2** are 1.04 eV and 3.34 eV, respectively. From the results, it can be seen that the HOMO orbitals of compound **2** are slightly higher than those of compound **1**, which implies that the internal electron jumping ability of compound **2** is stronger than that of compound **1**, and the output performance will be higher, so we speculate that the output performance of **2**-TENG will be slightly better than that of **1**-TENG. In addition to the electrode materials that have a greater influence on the output performance of the TENG, the contact tightness between the electrode materials also has an influence on the output performance. Therefore, we tested the two compounds under the same experimental conditions by taking the same amount of compounds **1** and **2** in the same mortar for the same amount of time and grinding them into powder and then uniformly coating them on copper tape to minimize the influence of other errors on the output performance of the TENG.

The working principle of TENGs based on MOF materials is based on the coupling of frictional charging and electrostatic induction. The simplest vertical contact–separation model was used in this experiment (Appendix A). The positive electrode materials of the friction power generator are compounds **1** and **2**, and the negative electrode material is polyvinylidene difluoride (PVDF), named **1**-TENG and **2**-TENG.

The output performance of the TENG was tested as shown in Figure 4, with short-circuit currents and open-circuit voltages at 5 Hz of 40.4 μA and 406.4 V, and 80.5 μA and 543.8 V, respectively (Figure 4a); with respect to the Isc and Voc of **1**-TENG, the Isc and Voc of **2**-TENG are larger. At 5 Hz, the values obtained with **1**- and **2**-TENG through the rectifier bridge are 39.0 μA and 76.8 μA, respectively (Appendix A); the charge densities are 87.0 μC m^−2^ and 101.2 μC m^−2^, respectively, based on the power densities and short-circuit currents of the friction nanogenerator at different load resistances; when the load resistance was 10 MΩ, the instantaneous power peaked at 1455.1 mW m^−2^ and 2629.7 mW m^−2^, respectively (Figure 4b). We also tested the output performance of compounds **1** and **2** under different frequency conditions. It was found that Isc (Figure 4c) and Voc (Figure 4d) increased with increasing frequency, and the output performance of **2**-TENG was higher than that of **1**-TENG at 1–8 Hz. When **2**-TENG was operated at 8 Hz, the current and voltage could reach 121.7 μA and 568.9 V. The test results showed that the magnitude of the compound’s output performance was **2**-TENG > **1**-TENG, which indicates that the more F is contained, the stronger the overall electronegativity of the moiety and the obvious electron withdrawal effect is, and the greater the TENG output performance of the compounds is. The specific output performance data of **1**-TENG and **2**-TENG are shown in Appendix A.

In addition, we tested the stability and durability of **2**-TENG at 5 Hz. The cycling test results showed that after 50,000 cycles, the Isc and Voc values of **2**-TENG did not show significant changes and remained in a stable state (Figure 5), and the Isc and Voc values of **1**-TENG did not show significant changes (Appendix A), which indicates that the samples have good stability and lays the groundwork for their practical applications. To further verify the stability of the friction electric materials, we observed the morphology of compounds **1**–**2** using scanning electron microscopy (SEM) and the elemental distributions of compounds **1** and **2** with characteristic K spectral lines (EDS) (Appendix A). The SEM results showed that the morphologies of compounds **1** and **2** were almost unchanged before and after the experiments, indicating that compounds **1** and **2** were relatively stable.

Metal corrosion protection has always been a hot topic of research and application because it can cause huge economic losses and serious accidents [33,34]. The TENG can harvest available energy from the surrounding environment to form a self-powered applied current cathodic protection system. A TENG-powered cathodic protection system is an effective means of electrochemical corrosion control, in which the oxidation reaction occurs mainly at the anode while inhibiting the corrosion of the cathode (i.e., the protected metal), which realizes the ideal mode of utilizing the energy of nature to protect metals from corrosion.

Based on the superior output performance of **2**-TENG, it can be used for self-powered cathodic protection. The schematic diagram of the anti-corrosion mechanism is shown in Figure 6a. The **2**-TENG output signal is converted from DC to be rectified by connecting a rectifier bridge, the positive pole of which is connected to a platinum electrode, and the negative pole of which is connected to carbon steel. Rust can allow a visualization of the degree of metal corrosion, and the degree of metal corrosion is usually judged by observing rust spots. In our experiments, we used 3.5 wt% NaCl in aqueous solution to simulate a seawater electrolyte and determined the effectiveness of protection by observing and comparing the surface morphology of the carbon steel with and without **2**-TENG connected. By comparing the rust spots on the surface of carbon steel after 0 h, 1 h, 3 h, 5 h, and 7 h, it was found that the surface of the carbon steel with **2**-TENG attached was almost free of rust spots in 0–7 h, but the surface of the carbon steel without **2**-TENG attached became more and more rusty with the extension of time (Figure 6e). This verifies that the corrosion protection method of self-powered cathodic protection is effective.

We evaluated the properties, corrosion degree, and protection efficiency of carbon steel using open-circuit potential (OCP), EIS, and Tafel with and without **2**-TENG connected. It was found that the open-circuit potential (OCP) of the protected metal carbon steel was stabilized at 0.20 V when no **2**-TENG was connected, whereas when **2**-TENG was connected, the OCP of the protected metal carbon steel decreased rapidly to −0.10 V. This was due to the negative migration of the OCP due to the charge transfer between the sample and the solution, and the negative migration indicated that the metal carbon steel was in the protected stage. When **2**-TENG is removed, the OCP value returns to the original position again (Figure 6b).

The size of the radius of the EIS circle in the impedance spectrum usually indicates the magnitude of the electrical resistance in the electrochemical reaction; the larger the radius of the circle, the more difficult it is for the electrochemical reaction to take place, which leaves the carbon steel unprotected and corroded (Figure 6c). The carbon steel with **2**-TENG attached shows a smaller semicircle than does the carbon steel without **2**-TENG attached, indicating that the internal resistance of the connection is smaller. This is due to the fact that the charge transfer between the sample and the solution makes the carbon steel with **2**-TENG connected more susceptible to electrochemical reactions than the carbon steel without a TENG connected. The equivalent circuit diagram in the corrosion study is clearer and more concise in explaining and illustrating the EIS plot, which is located in the upper-left corner of the EIS plot. (Rs: solution resistance; Rct: charge transfer resistance; QdI: double-layer resistance).

At the same time, we tested the polarization curves with and without the **2**-TENG carbon steel connected. From the experimental data, it can be seen that the polarization potential (Ecorr) of the carbon steel with **2**-TENG connected was negatively shifted compared with that of the carbon steel without **2**-TENG connected (Figure 6d), which is consistent with the test results of the OCP. In addition, the polarization current (Icorr) of the carbon steel with **2**-TENG attached was slightly higher than the value without **2**-TENG attached, which was attributed to the higher number of electrons generated by **2**-TENG on the surface of the carbon steel, which limited the electrochemical reaction. In summary, **2**-TENG can effectively protect carbon steel cathodes.

## 3. Discussion

The electronegativity of the triboelectric nanogenerator electrode material affects the TENG’s output performance when the overall electronegativity of the groups is strong, i.e., the more significant electron-absorbing effect, the stronger the TENG output performance of the compound. In this paper, more electronegative groups are introduced into the terminal microstructure to improve the TENG’s output performance under the control of a similar spatial structure. The experimental results show that **2**-TENGs with more F materials show superior output performance, and the materials are rich in electronegative F, which realizes electron delocalization conjugation and is more conducive to electron gain and loss. In practical applications, **2**-TENG can be used as a self-powered cathodic protector to protect metallic materials from corrosion based on its high stability and excellent output performance. Therefore, this study not only reveals the effect of terminal microstructure negativity on the output performance of TENGs, but also provides a simple method for the design of novel self-powered cathodic protection materials in the future.

## 4. Materials and Methods

### 4.1. Compounds Preparation

All the chemicals and reagents used in this experiment were obtained by purchasing them through Aladdin’s Reagent platforms and were used directly without further purification. Compound **1** was prepared on the basis of reports in the literature [35] and compound **2** was made through our own exploration. The experimental data for the MOF-based TENGs of compounds **1**–**2** were collected with the relevant experimental apparatus. Electrochemical measurements were carried out on carbon steel with and without **2**-TENGs attached via a three-electrode system on an electrochemical workstation. The crystal structures were determined with a X-ray single-crystal diffractometer and solved, and the single-crystal structures were plotted with Diamond 4 software. Thermogravimetric analysis (TGA) data were collected on a Netzsch STA 449C thermal analyzer. Compounds were subjected to the Mott—Shottky test with a three-electrode system on an electrochemical workstation (CHI 660E, Shanghai Chenhua Instrument Co., Ltd., Shanghai, China). The chemical bond determination of the compounds was tested via FT-IR; purity was characterized via powder X-ray diffraction mapping (PXRD) under Cu-kα irradiation using an instrument manufactured by Bruker D8 Advance. For testing the elemental composition and valence states, XPS was used to obtain spectral data plots of the compounds using Al-Kα as the ray source. The morphology, elemental composition, and distribution of the compounds were tested with a German-made FE-SEM and its accompanying EDS. The Isc and Voc of the friction nanogenerators were tested on an SR570 low-noise current amplifier (Stanford Research Systems) and a 2657A high-power system source meter. The bandgap size of the samples was tested with a UV-2600 UV-Vis (Manufacturer Perkin Elmer Instruments (Shanghai) Co., Shanghai, China, Model Lambda 950) diffuse reflectance spectrometer.

### 4.2. Synthesis of [Co (2,3,5,6-Tetrafluoroterephthalic acid)] (Compound ***1***)

A solid mixture of Co(NO_3_)_2_-6H_2_O (29.1 mg, 0.1 mmol) and 2,3,5,6-tetrafluoroterephthalic acid (23.8 mg, 0.1 mmol) was weighed and dissolved in a mixture of DMF (2 mL), water (2 mL), and ethanol (6 mL), the resultant solution was transferred to a small glass vial with a pipette gun, and the vial was opened and allowed to evaporate naturally at room temperature; light-purple crystals were precipitated after three weeks, washed three times with ethanol, and dried at room temperature.

### 4.3. Synthesis of [Co (2,5-Bis(trifluoromethyl)terephthalic acid)] (Compound ***2***)

A solid mixture of Co(NO_3_)_2_-6H_2_O (15.2 mg, 0.05 mmol) and 2,5-bis(trifluoromethyl)terephthalic acid (15.2 mg, 0.05 mmol) was weighed and dissolved in a mixture of DMF (6 mL), water (2 mL), and ethanol (2 mL), the resulting solution was transferred to the reactor using a pipette gun, and then the reactor was placed in an oven preheated to 80 °C for 24 h. The solution was cooled to room temperature, washed three times with DMF and placed in a vacuum drying oven for drying.

### 4.4. Preparation of 1/2 Base TENG

With regards to positive electrode sheet preparation for the TENG, compound **1** and **2** were ground, evenly coated on a 5 cm × 6 cm copper sheet, and is then coated with the conductive silver epoxy resin of the copper wire fixed on the other side of the copper sheet. The fixed copper wire of the copper tape surface was covered with a layer of transparent adhesive tape, and finally cut out into 5 cm × 5 cm sections.

In terms of TENG negative electrode sheet (PVDF) preparation, we weighed the appropriate amount of PVDF powder, dissolved in a mixture of DMA and acetone, stirring at 60 °C until the PVDF powder was completely dissolved to stop the heating. It was cooled to room temperature, and the desktop homogenizer was uniformly coated with the PVDF mixture on the Kapton membrane. The membrane was spin-coated, placed in the oven for drying at 80 °C for 4 h, and then removed to be cool to room temperature. A copper tape measuring 5 cm × 6 cm was applied to the reverse side of the PVDF film, and the negative electrode sheet was prepared in the same way as the positive electrode sheet was.

## Figures and Tables

**Figure 1 molecules-28-07894-f001:**
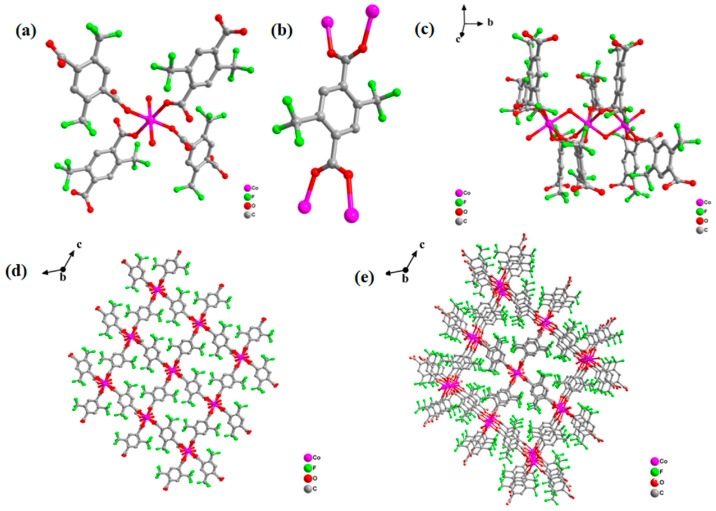
(**a**) Pattern diagram of the coordination environment of the Co metal center; (**b**) pattern diagram of the coordination environment of the organic ligand 2,5-bis(trifluoromethyl)terephthalic acid; (**c**) ligand chain fragment of the Co(II) octahedron; (**d**) planar structure of the compounds shown along the a-axis; and (**e**) three-dimensional mesh structure of the compounds. C, grey; O, red; Co, rosy red; all H atoms are omitted for clarity.

**Figure 2 molecules-28-07894-f002:**
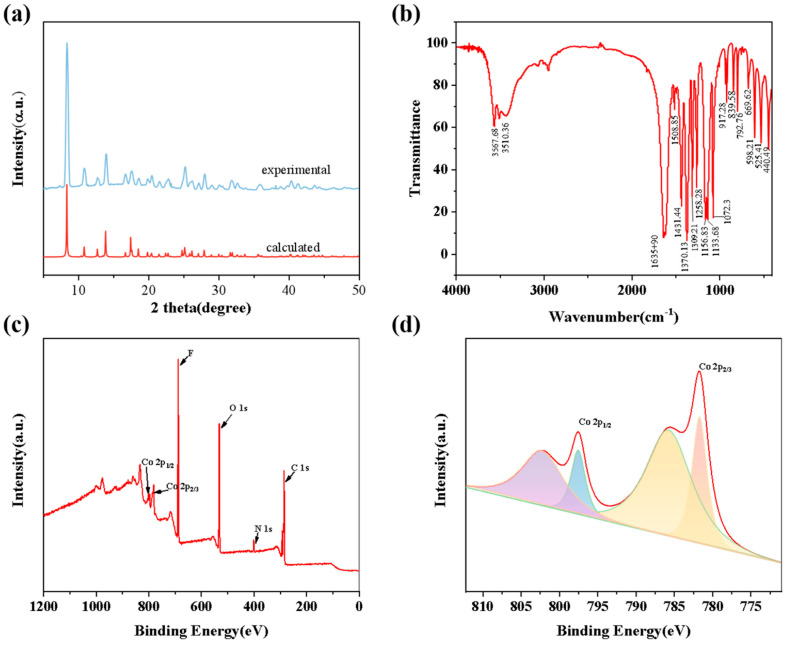
(**a**) XRD; (**b**) FT-IR; (**c**) XPS full spectrum; (**d**) Co ion XPS profile of compound **2**.

**Figure 3 molecules-28-07894-f003:**
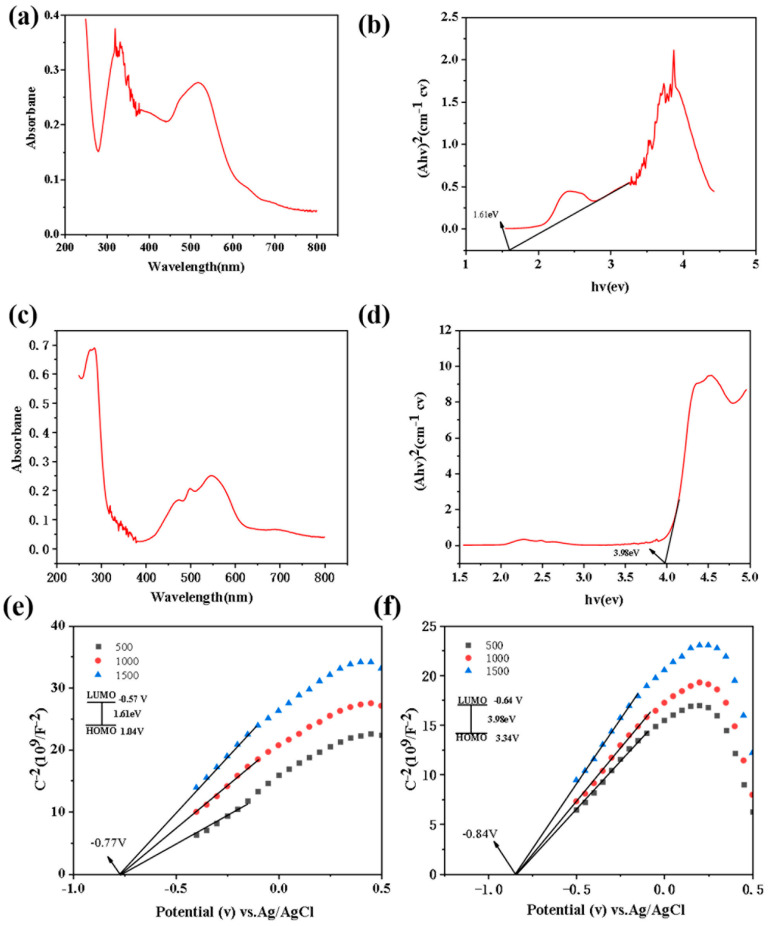
(**a**,**c**) UV-Vis diffuse reflectance spectra of compounds **1**–**2**; (**b**,**d**) the corresponding Tauc plots for a,c, respectively; (**e**,**f**) Mott—Schottky curve of compounds **1**–**2** tested in 0.2 M Na_2_SO_4_ solution.

**Figure 4 molecules-28-07894-f004:**
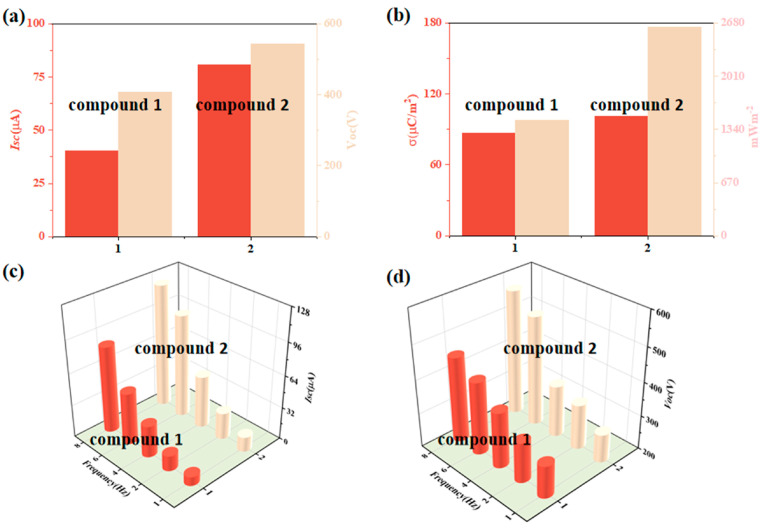
Comparison of (**a**) current–voltage, (**b**) charge power density, (**c**) different Hz currents, and (**d**) different Hz voltages for compounds **1**–**2**.

**Figure 5 molecules-28-07894-f005:**
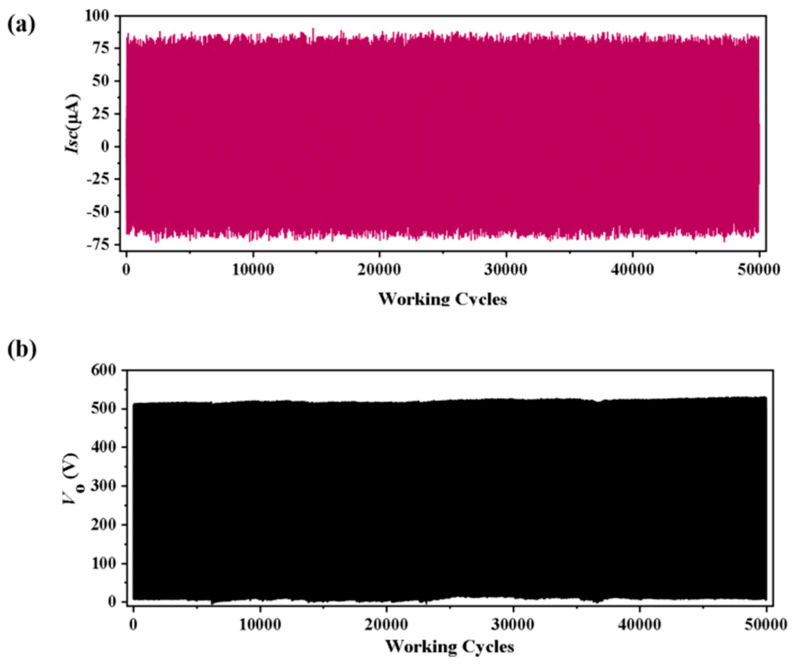
2−TENG after 50,000 cycles of (**a**) Isc; (**b**) Voc.

**Figure 6 molecules-28-07894-f006:**
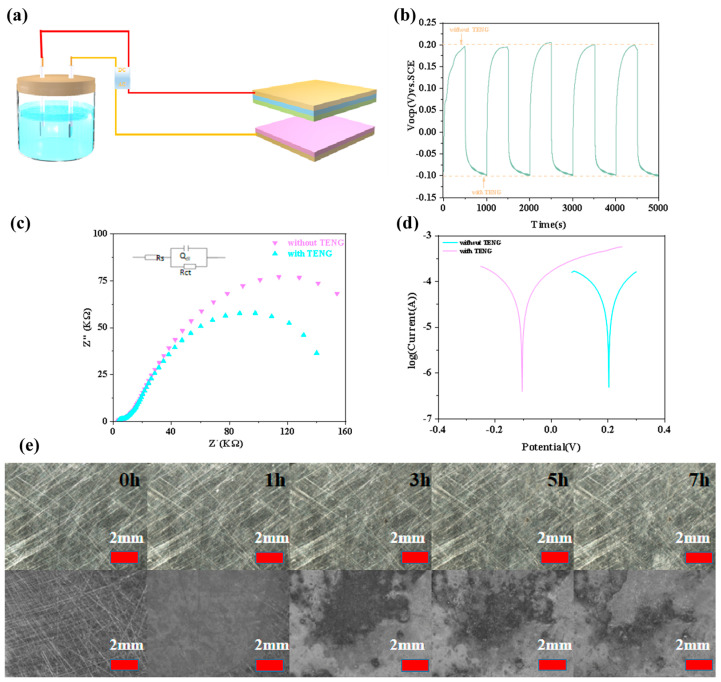
(**a**) Mechanism diagram of cathodic protection of Q235 carbon steel by 2-TENG. (**b**) OCP and 5−cycle stabilization. (**c**) EIS and equivalent circuit diagram (inset). (**d**) Tafel of carbon steel with and without 2−TENG attached. (**e**) Metallographic micrographs of carbon steel with and without 2-TENG attached after immersion in 3.5 wt% NaCl solution for 0, 1, 3, 5, and 7 h.

## Data Availability

Data are contained within the article and Appendix A.

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
