# Peer review of "Multi-F-Structured MOF Materials Enhance Nanogenerator Output Performance for Corrosion Protection of Metallic Materials"

_molecules, 2023, doi:10.3390/molecules28237894_

Round 1

Reviewer 1 Report

Comments and Suggestions for Authors

This manuscript, entitled "Multi-F-structured MOF Materials Enhance Nanogenerator Output Performance for Corrosion Protection of Metallic Materials," is well-written and informative. It presents interesting and important findings on the use of MOF materials in triboelectric nanogenerators. The authors, Jia-Bin Xiong et al., have clearly demonstrated that the modulation of the end microstructures of MOFs using the difference in F content can significantly improve the output performance of friction power generation. Additionally, the obtained polyacidic ligand materials can be used to construct self-powered corrosion protection systems, effectively shielding metallic materials from corrosion.

Overall, this manuscript is a valuable contribution to the field of triboelectric nanogenerators and MOF materials. I recommend it for publication in “molecules”, with the following minor corrections.

·         The authors could discuss the potential applications of their findings in more detail. For example, they could mention how their self-powered corrosion protection systems could be employed to safeguard metallic structures in harsh environments, such as bridges and pipelines.

·         Furthermore, the authors should proofread the manuscript carefully to correct any minor grammatical errors.

In conclusion, this is a well-written and informative manuscript that presents interesting and important findings. I recommend it for publication in your MDPI's journal "Molecules."

Comments on the Quality of English Language

minor grammatical proofreading is required 

Reviewer 2 Report

Comments and Suggestions for Authors

This manuscript explores the modulation of the electronegative groups in MOFs on triboelectric nanogenerator output performance. There are several comments that must be addressed to improve the quality of this manuscript before it is in an acceptable condition for publication:

1. Discussions on the crystal structure of compound 1 needs to be provided in section 3. Likewise, the PXRD of compound 1 is missing.

2. My main concern is on the particle size distributions of compounds 1 and 2 after grinding. Are they similar, and if not can the difference in surface areas have an influence on the output performance?

3. More information needs to be provided in the experimental method in section 2. For example, “purchasing through commercial platforms” is too vague. Which suppliers were used? Citations should also be provided on the preparation methods, especially when it was stated that the compounds were “prepared on the basis of literature reports”. It was also stated that “light purple crystals were precipitated after three Wednesdays”. Does this mean three weeks?

4. In Figure 2d, what do the peaks at around 786.0 eV and 802.5 eV represent?

5. In lines 133-134, it was stated “Compounds 1 and 2 show significant weight loss at 110 and 223 degrees Celsius, respectively, which is the result of volatilization of water molecules from the compounds”. I think a weight loss at 223 degrees Celsius would more likely be DMF than water.

6. Higher quality figures are needed. In particular, the numbers are too crowded in Figure 2b, and the insets of Figures 3a and 3c cannot be read clearly.

7. What does “σ” in the Introduction (line 35) refer to?

8. There are several grammatical errors/typos that needs to be fixed. For example: “flexible customizability of crystalline materials” (line 42), “of is” (line 53), “the effect of the effect of” (line 59), “pre The” (line 93), “performance. performance.” (line 162), “the harder the electrochemical reaction occurs” (line 244).

Comments on the Quality of English Language

Please fix the grammatical errors and typos as mentioned in the comments.

Reviewer 3 Report

Comments and Suggestions for Authors

I have reviewed an article entitled “Multi-F-structured MOF materials enhance nanogenerator output performance for corrosion protection of metallic materials”, written by Jia-Bin Xiong and co-workers.

In this work, the authors study the preparation and characterization of two Co(II) derivatives of ligands 2,5-bis(trifluoromethyl)terephthalic acid (compound 2) and 2,3,5,6-tetrafluoroterephthalic acid (compound 1). The authors reported the application of two MOFs materials to triboelectric nanogenerator. They study the output performance of friction power generation in function of F content.

In general, the text is full of typographical errors, the figures are of poor quality and the authors do not provide a detailed discussion of the results obtained, nor do they compare them with others works. In the experimental and results sections, only three references appear and one of them [28] is missing in the text.

Comments and Suggestions for authors:

In my opinion, this article does not have sufficient quality to be published in the journal Molecules. I would recommend a thorough review of the bibliography, improving the quality of the figures presented in the text, expanding the discussion of the results by comparing them with other published ones and correcting all errors found.

My main comments are listed below:

-                      Lines 56-57: “… ligands (2,5-bis(trifluoromethyl)terephthalic acid and 2,3,5,6-tetrafluoroterephthalic 56 acid), as friction electrode materials (referred to as compounds 1 and 2, respectively)”. It is an error and It would be just the opposite

-                      Compounds preparation: Lines 72-74: “Compound 1 was prepared on the basis of literature reports of modification and compound 2 was prepared on the basis of literature reports”  What are those reported jobs? no previous work is recorded.

-                      Synthesis of compound 1 and 2. Lines 80 and 88. “Cu”? It is an error, the compounds are two derivatives of Co(II). Lines 91-93: “and the 91 resultant solution was transferred to a reactor using a pipette gun, which was 92 placed in a pre.” I don't understand this sentence, I think it is incomplete

-                      Results and discussion. The authors describe the structure of compound 2 but absolutely nothing is said about the structure of compound 1. Furthermore, the molecular formula of neither of the two compounds appears.

-                      Lines 115-116: “c, grey; o, red; co, rosy red”. Please write the chemical elements correctly.

-                      Lines 117-119: “The XRD patterns of the compounds were in agreement with the previously reported PXRD patterns (Figure. 2a) both fitted well, indicating good crystallinity and purity of the compounds“. What are the references of the previously reported PXRD?. In addition, the PXRD pattern of compound 1 is missing in the document and in the SI.

-                      In the figure 2c the authors shown a panoramic XPS spectrum for compound 2. The signals of the elements C1s, O1s, F1s and Co2p are shown and the signal of N1s is shown. Have the authors verified that the compound is not contaminated with traces of the starting reagent or has the presence of DMF?

-                      In the thermogravimetric study, the authors do not discuss in the text whether there is a good agreement between the experimental data and the calculated % mass loss values if they correspond to the number of water molecules with which the compounds crystallize.

-                      Lines 132 and 133: at 70 degrees Celsius .. and …. at 110 and 223 degrees Celsius are errors, it is more correct to say 70 °C or 110 and 223 °C.

-          Line 162: the word “performance” is repeated in the sentence.

-          It is impossible to see the data in figures 2, 3, 4 and 6 due to the small size of the figures and their poor quality.

Comments on the Quality of English Language     The authors should further review the text to eliminate any errors found

Reviewer 4 Report

Comments and Suggestions for Authors

This study by Xiong et al presents two MOFs with similar spatial configurations but different fluorine (F) contents, demonstrating that higher F content enhances the electrical output of TENGs. These findings suggest potential for MOFs in creating self-powered systems for effective corrosion protection of metals. This work is interesting, but not well organized and characterized. Thus, it was suggested to accept after addressing the list of issues.

1.      For the claimed single crystals, the corresponding CCDC numbers should be provided along with complete crystallographic tables in the Supplementary Information.

2.      Terminology such as Multi-F-structured, MOF, and F content must be fully named upon their initial mention.

3.      The PXRD data for compound 1 should be included.

4.      The N2 adsorption isotherms, currently not reflecting adequate activation of the MOF pores, should follow the activation procedure described in the literature and be re-collected, necessitating a revision of the related section and Figure S3c-d. Check the paper for proper activation: CrystEngComm,2013,15, 9258; Chem. Mater. 2017, 29, 1, 26–39.

5.      Thermal Gravimetric Analysis (TGA) should not indicate water loss if samples are correctly activated, and the noted weight losses at specific temperatures could indicate the need for additional solvent washing steps prior to activation.

6.      Clarification is needed on the weight percentage increase from 300-800°C in Figure S3, including temperatures related to isotherms.

7.      Figure 3 requires a more detailed caption and a reordering of subfigures 3b and 3c, with an elaboration on UV results and a prelude to the Mott-Schottky analysis.

8.      In Figure 4, labels for compounds 1 and 2 should be added for clarity.

9.      Section 1.2 in the Supplementary Information has typographical errors, including an incorrect reference to "three Wednesdays," which should be corrected.

10.   The literature cited needs to be updated.

Comments on the Quality of English Language

The manuscript requires a clear and thorough explanation of the characterization findings.

Round 2

Reviewer 2 Report

Comments and Suggestions for Authors

The manuscript has been improved.

Comments on the Quality of English Language

The grammatical errors noted in my previous comment have been fixed.

Author Response

As per the reviewer, there are no changes needed, thanks to the reviewer for recognising this!

Reviewer 3 Report

Comments and Suggestions for Authors

The authors have included all the suggestions made in the manuscript, but in the Supplementary Information has typographical errors in 1.2 Synthesis of compounds 1-2 part. Is Cu or Co?

In the other hand, the Figure S12 shows the experimental X-ray diffraction pattern of compound 1, in the range 19-23 2Theta shows the presence of additional peaks not present in the calculated diffraction pattern, therefore the sample is doped. Have the authors taken this into account in the study?
